# Operational priorities for engaging with India's private healthcare sector for the control of tuberculosis: a modelling study

Saskia Ricks [1], Ananya Singh,[2] Ridhima Sodhi [2], Arnab Pal,[2]
Nimalan Arinaminpathy [1]

[1]Imperial College London School of Public Health, London, UK
[2]Clinton Health Access Initiative, New Delhi, India

**Correspondence to**
Dr Saskia Ricks;
saskia.ricks12@imperial.ac.uk

## ABSTRACT

**Objectives** To estimate the potential impact of expanding services offered by the Joint Effort for Elimination of Tuberculosis (JEET), the largest private sector engagement initiative for tuberculosis (TB) in India.

**Design** We developed a mathematical model of TB transmission dynamics, coupled with a cost model.

**Setting** Ahmedabad and New Delhi, two cities with contrasting levels of JEET coverage.

**Participants** Estimated patients with TB in Ahmedabad and New Delhi.

**Interventions** We investigated the epidemiological impact of expanding three different public–private support agency (PPSA) services: provider recruitment, uptake of cartridge-based nucleic acid amplification tests and uptake of adherence support mechanisms (specifically government supplied fixed-dose combination drugs), all compared with a continuation of current TB services.

**Results** Our results suggest that in Delhi, increasing the use of adherence support mechanisms among private providers should be prioritised, having the lowest incremental cost-per-case-averted between 2020 and 2035 of US$170 000 (US$110 000–US$310 000). Likewise in Ahmedabad, increasing provider recruitment should be prioritised, having the lowest incremental cost-per-case averted of US$18 000 (US$12 000–US$29 000).

**Conclusion** Results illustrate how intervention priorities may vary in different settings across India, depending on local conditions, and the existing degree of uptake of PPSA services. Modelling can be a useful tool for identifying these priorities for any given setting.

## STRENGTHS AND LIMITATIONS OF THIS STUDY

⇒ We have used detailed programmatic data to inform costs, effectiveness and real-world coverage of the different interventions discussed in this analysis to help identify operational priorities.

⇒ However, due to a lack of city-specific data for Delhi and Ahmedabad, the mathematical model was instead calibrated to prevalence and infection surveys from Chennai, another large Indian city.

⇒ Due to data gaps in healthcare provider behaviour, we made assumptions informed partly by expert opinion, to estimate how this behaviour is likely to change through engagement efforts.

with this sector[6–10]: not only in notifying TB to public health authorities, but also in offering high-quality diagnostics and treatment adherence support for patients being managed by private providers. Early efforts to engage effectively with the private sector faced substantial challenges, principally arising from a lack of trust.[11] Despite TB being made a notifiable disease in 2012 in India,[12] few private providers reported their patients with TB to public health authorities. However, from 2013 onwards, pilots of 'public–private support agencies' (PPSAs) showed how private providers could be successfully engaged, through supporting private providers to notify TB, as well as making high-quality diagnostics and treatment support available for patients being managed by the private sector.[7 9]

To take the PPSA model to scale across the country, the Joint Effort for Elimination of Tuberculosis (JEET) was launched by three partner organisations (the Clinton Health Access Initiative (CHAI); Centre for Health, Research and Innovation and Foundation for Innovative New Diagnostics) in 2018.[13] Supported by the Global Fund and coordinated by the Central TB Division of India, it

## INTRODUCTION

Tuberculosis (TB) is an airborne infectious disease[1] and a major cause of death worldwide, responsible for 10.6 million cases and 1.6 million deaths in 2021.[2] India accounts for over a quarter of global TB burden, with 2.9 million cases in 2021.[2] A major challenge in India is the presence of a vast, fragmented private healthcare sector, which manages the majority of patients with TB in the country.[3–5] The past decade has seen increasing recognition of the importance of engaging effectively

is the largest private sector engagement initiative for TB in India to date.[14] The main activities include increasing case notifications by engaging with private providers, improving the quality of TB diagnosis by providing free access to molecular diagnostics, and improving treatment outcomes by providing free fixed dose combination (FDCs) and adherence support.[13] By 2020, JEET was operating in 485 cities across India[13]; these developments accompanied a substantial increase in notifications from private providers, accounting in 2022 for around 30% of total TB notifications in India.[15]

Despite these achievements, there remains much ground to be covered. Moreover, progress in PPSA implementation has varied across the country, with greater uptake of available services in some cities than in others. One important question is what kinds of PPSA services should be promoted in any given setting, depending on what levels of uptake have already been achieved. In particular, in a given city, should future efforts for strengthening PPSA efforts place greater priority on increasing the uptake of molecular diagnostics, or of adherence support mechanisms? Here, we aimed to address this question using mathematical models of TB transmission, to identify the most cost-effective strategies for future PPSA expansion.

## METHOD
### Model overview
As two examples, we focused on the urban settings, Ahmedabad and Delhi, with contrasting levels of uptake of PPSA services. Ahmedabad has population of around 7.7 million and is the largest city in the Indian state of Gujarat; Delhi has a population of around 33 million and is the capital city of the country. PPSA operations in both cities began in 2018[16] and have followed different trajectories with, for example, services in Delhi seeing substantially less uptake of FDCs than Ahmedabad.

We developed a deterministic, compartmental model of TB transmission among adults consistent with an urban setting in India (figure 1). The model captures routine TB services, including the scale-up of the Revised National TB Control Programme between 1997 and 2007,[17] and the scale-up of private sector engagement

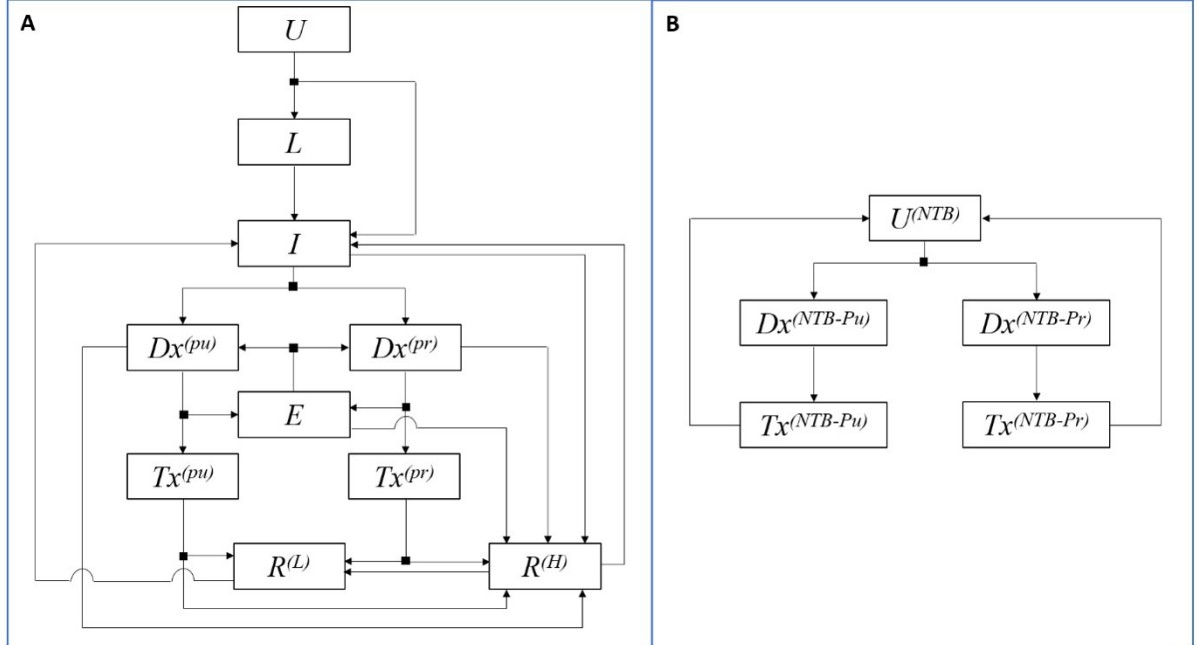

**Figure 1** Schematic illustration of the model structure. (A) The TB transmission model structure. Model compartments are as follows: uninfected with TB (U), latent infection (L), active TB disease (I), presented for care and awaiting diagnosis (Dx), on TB treatment (Tx), temporarily dropped out of the TB care cascade due to missed diagnosis or pretreatment loss to follow-up (E) and recovered with a low risk of relapse (R(L)) or with a high risk of relapse (R(H)). Dx and Tx are stratified by healthcare sector: private (pr) and public (pu) sector. The private sector is further subdivided into several types of providers as described in the main text. Not shown for simplicity is the acquisition and transmission of DR-TB. See (online supplemental information) for further technical details, including model equations and calibration. (B) The non-infected TB-symptomatic model structure. Model compartments are as follows: uninfected with TB but TB symptomatic (U(NTB)), presented for care and awaiting diagnosis in the public sector (Dx(NTB-Pu)) or private sector (Dx(NTB-Pr)), and TB treatment after a false-positive diagnosis in the public sector (Tx(NTB-Pu)) or private sector (Tx(NTB-Pr)). The private sector is further subdivided into several types of providers as described in the main text. In order to maintain a constant population size among the non-infected TB-symptomatic population, we assume that on treatment completion, individuals return to the uninfected TB-symptomatic state. The size of this population was determined so that among all patients presenting for care, there is a 10% prevalence of TB, in either the public or private sector.[19–21] DR-TB, drug-resistant TB; TB, tuberculosis.

**Table 1** Calibration targets used to estimate model parameters

| Indicator | Value | Source |
|---|---|---|
| Annual risk of TB infection | 2% (1%–3%) | [22] |
| TB prevalence | 259 per 100 000 (217–299) | [23] |
| TB mortality | 32 per 100 000 (30–34) | [33] |
| Per cent of TB cases that are MDR | 4% (3%–5%) | [33] |

We assumed the same prevalence, annual risk of TB infection, TB mortality and per cent of TB cases that are MDR across Ahmedabad and Delhi.
MDR, multidrug resistant ; TB, tuberculosis.

from 2017 onwards.[18] We incorporated the acquisition and transmission of drug-resistant TB (DR-TB), including both rifampicin-resistant and multidrug-resistant TB (MDR-TB). For simplicity, we ignored HIV status, and the distinction between pulmonary and extrapulmonary TB, although we conducted a sensitivity analysis on the latter. We separately simulated a non-TB symptomatic population (figure 1) to count the overall number of TB diagnoses and false positive treatments among presumptives mistakenly diagnosed as having TB. This population has no influence on the 'main' model of TB transmission dynamics, and its purpose is only to quantify volumes of diagnostic tests and treatments. As implemented in previous work, we calibrated the size of the non-TB symptomatic population to give a 10% prevalence of TB among those presenting for care, in either the public or private sector.[19–21] Online supplemental information provides further details on the model, including governing equations and parameter values (online supplemental tables S1–S7), and a breakdown of the proportion of patients visiting different providers for diagnosis and treatment (online supplemental tables S8, S9, respectively).

### Data sources
To reflect a typical urban setting in India (such as Ahmedabad and Delhi), we used survey data of adults (>15 years old) for the annual risk of TB infection and the prevalence of TB.[22 23] Table 1 summarises the data from these sources. Online supplemental tables S2–S5 list parameter values used in the model (TB natural history, diagnosis, treatment and costs, respectively).

We incorporated uncertainty in parameter values and model inputs through Adaptive Bayesian Markov chain Monte Carlo.[24] Thus, the uncertainty in model projections reflects the uncertainty in the model inputs; we refer to the uncertainty in model projections as the Bayesian credible intervals, using the 2.5th and 97.5th percentiles to reflect the lower and upper bound of an interval.

We also drew a range of operational data from CHAI, to inform elements such as existing uptake of JEET services in Ahmedabad and Delhi; provider behaviour over time in response to CHAI engagement efforts; and treatment outcomes under different types of adherence support

in the private sector. Further details are provided in the 'Modelling existing PPSA services' section.

### Care cascade
We assumed that on development of active disease, there is a delay (determined by calibration to the available data) before individuals seek care.[10 25] We modelled public and private providers separately, assuming: (1) the quality of diagnosis to be lower in the latter than the former and (2) once individuals are diagnosed with TB and initiated on treatment, a lower rate of treatment completion in the private sector, due to the general lack of adherence support.[26 27] Owing to a lack of systematic data for the private sector when not engaged with a PPSA, we allowed for a wide range of parameter values relating to both diagnosis and treatment in this sector (online supplemental table S4). Online supplemental information also provides further details on the non-TB symptomatic population care cascade.

### Modelling existing PPSA services
#### Provider recruitment
When commencing operations in any new setting, JEET conducts provider mapping to identify the highest-volume providers that should be prioritised. 'Engagement' with providers is achieved by field officers to encourage providers to notify their TB cases. However, even after these efforts, not all engaged providers notify TB, and require further engagement by field officers to do so, coupled with a strong enforcement of regulations. Accordingly, we distinguished different types of private providers: those who are not engaged; those who are engaged but not notifying TB ('inactive' providers); and those who are notifying TB ('active' providers) (online supplemental table S10). We define 'active' providers as those notifying TB at least once a month. Active providers may, over time, turn inactive, for three main reasons: first, some providers may not have a sufficient number of patients with TB every month and thus may notify infrequently depending on when they see a new patient with TB; second, some providers, especially those that handle a smaller volume of patients with TB, are seen less regularly by the field team and are therefore less likely to notify and third, some providers may be dissatisfied with the services offered by JEET, and therefore, intentionally stop notifying patients. If 'active' providers are defined as those notifying TB at least once a month, we accounted for this rate of attrition by weighting the amount of CB-NAAT (cartridge-based nucleic acid amplification testing) and FDC issued by the number of months providers are active for.

The transmission model follows the patient perspective of the careseeking journey; an important parameter relating to PPSA activities is the probability that a patient, when seeking care for their symptoms, visits a private provider among the three categories listed above. To estimate these probabilities, we modelled a 'heterogeneity

**Table 2** CB-NAAT use among active private providers in Ahmedabad and Delhi in 2019

| | City | |
| --- | --- | --- |
| | **Ahmedabad** | **Delhi** |
| Total no of active providers | 762 | 894 |
| No of active providers that always use CB-NAAT | 65 | 35 |
| No of active providers that occasionally use CB-NAAT | 278 | 265 |
| No of active providers that never use CB-NAAT | 419 | 594 |
| Per cent of notifications that are CB-NAAT confirmed by occasional users | 22 | 15 |

To be classified as occasionally using CB-NAAT, we assumed that providers use CB-NAAT for at least 1% of notifications and less than 100% of notifications. Data are provided by CHAI.
CB-NAAT, cartridge-based nucleic acid amplification testing; CHAI, Clinton Health Access Initiative.

index' $\alpha$, such that (online supplemental figure S1 and table S11):

$$\text{Proportion of patients visiting an engaged provider} = (\text{Proportion of providers engaged})^{\alpha}$$

Thus, values of $\alpha < 1$ correspond with the situation where high-volume providers are preferentially engaged. We estimated the value of $\alpha$ using data provided by CHAI for both Ahmedabad and Delhi.

### TB diagnosis

Field officers engage with active providers to encourage them to offer CB-NAAT to their presumptive patients with TB. Specimen collection and transportation (SCT) agents collect the sputum samples from the provider clinic and transport them to laboratories where CB-NAAT testing is performed for free. While efforts are made to encourage active providers to use these services, the services are not mandated under Project JEET, in the interest of a collaborative relationship.

There is a multimodal distribution for the uptake of this CB-NAAT service among active private providers, including those who never use it; those who use it occasionally; and those for whom all notifications are CB-NAAT confirmed (online supplemental figure S2). In 2019, 55% and 66% of active providers in Ahmedabad and Delhi did not use CB-NAAT confirmation when notifying TB. Providers who do not confirm a TB notification with CB-NAAT instead evaluate patients using chest X-ray

and clinical judgement.[18 19 28] We assumed that inactive and unengaged private providers diagnose TB in the same way. We drew from the literature for the sensitivity and specificity of X-ray and clinical judgement in diagnosing TB[29 30] (online supplemental table S3).

Motivated by online supplemental figure S2, we divided active providers into three categories depending on their use of CB-NAAT: providers who always confirm TB notifications with CB-NAAT, providers who occasionally use CB-NAAT and providers who never use CB-NAAT. Table 2 shows the number of active providers in each of these categories for Ahmedabad and Delhi in 2019. Once active providers become a CB-NAAT user, field officers continue visits to encourage these providers to continue confirming TB diagnoses with CB-NAAT.

### TB treatment

JEET offers two different adherence services for TB treatment via treatment coordinators: government supplied FDCs and adherence support (table 3). Government supplied FDCs are supplied for free to the patient by the government, whereas patients on adherence support receive regular text messages and calls to ensure treatment adherence. Field officers visit active providers regularly to ensure they are continuing to recommend government supplied FDCs and adherence support.

**Table 3** Number of active private providers that offer government supplied FDCs and adherence support in Ahmedabad and Delhi in 2019

| | City | |
| --- | --- | --- |
| | **Ahmedabad** | **Delhi** |
| Total no of active providers | 762 | 894 |
| No of active providers that offer adherence support | 712 | 835 |
| Per cent of patients who agree to adherence support | 92 | 92 |
| No of active providers that agree to government supplied FDCs | 529 | 242 |
| Per cent of active providers that consistently use FDCs, unweighted | 45 | 10 |
| Per cent of active providers that consistently use FDCs, weighted | 39 | 7 |

Data provided by CHAI, based on existing operations.
CHAI, Clinton Health Access Initiative; FDC, fixed-dose combination.

We considered four categories of active providers in the model: active providers that use government supplied FDCs and offer adherence support, those that use government supplied FDCs but do not offer adherence support, those that offer adherence support but do not use government supplied FDCs and those that offer neither service. We drew from data provided by CHAI to inform treatment completion rates associated with each of these categories, active providers offering both services having the highest rate of treatment completion and those offering neither having the lowest rate (online supplemental table S4). Finally, we assumed that inactive and unengaged providers do not offer adherence support and thus patients seeking treatment from these providers have the lowest treatment completion rates.

All patients diagnosed with DR-TB by a private provider are transferred to the public sector and initiated onto the second line treatment.

### Interventions

We focused on the impact of the following four interventions:

- Activate providers: increasing the number of active providers among those already engaged but not notifying TB.
- Promote CB-NAAT (occasional users): increasing CB-NAAT use among occasional CB-NAAT users.
- Promote CB-NAAT (non-users): encouraging active providers that never use CB-NAAT into occasionally using CB-NAAT.
- Promote adherence mechanisms: increasing the use of government supplied FDCs among active providers.

We did not examine the impact of increasing adherence support, as current uptake is already high (above 90%) in Ahmedabad and Delhi.

We modelled a 'theoretical maximum coverage' scenario where all providers are amenable to intervention. In practice, however, some providers may remain uninterested in taking up the services offered despite sustained engagement efforts by field officers. To reflect these limits, we also modelled a 'feasible maximum' scenario where some proportion of providers remain unresponsive to the interventions listed above. Informed by expert consultation with CHAI, we assumed that: all inactive providers and occasional CB-NAAT users are interested in notifying and increasing their CB-NAAT use, whereas 40% of non-CB-NAAT users are uninterested in using CB-NAAT, and 35% of non-government supplied FDC users are uninterested in using government supplied FDCs (see online supplemental figure S3 for more detail).

For both cities, we examined the epidemiological impact of increasing the four activities to their feasible and theoretical maxima, compared with an indefinite continuation of current levels of routine (publicly offered) TB and PPSA services. We modelled the scale-up of existing PPSA activities from 2017 to current levels in 2020. We assumed the interventions are scaled up linearly between 2020 and 2021, and simulated their impact until 2035.

### Economic evaluation

We took the programmatic perspective, that is, the costs that the national TB programme would have to pay. To examine which PPSA service should be prioritised in the future, we calculated the cost of expanding the different services to their feasible and theoretical maximum using operational cost data from 2019 for Ahmedabad and Delhi. Online supplemental table S12 summarises the total cost in 2019 for the different activities involved in the service provisions offered by JEET. These activities include the salaries of the field officer, treatment coordinators, hub agents and SCT agents; the cost of continuing medical education (seminars organised for providers to increase engagement); the cost of promotional materials which includes the cost of designing, printing and dissemination; the cost of transporting sputum samples to the laboratories, (including the cost of procuring falcon tubes, carrier boxes, masks and hand sanitiser); the cost of CB-NAAT tests, including test cartridges and manpower; and the cost of government supplied FDCs. We ignored sunk costs (office expenditure and state programme management costs) that have already been incurred in Ahmedabad and Delhi, and thus would not influence decisions for future interventions.

We categorised each cost item listed into online supplemental table S12 into six different categories: provider activation, provider retention, diagnostic engagement, diagnostic logistics, treatment engagement and treatment logistics (described in further detail in online supplemental table S13). Online supplemental table S14 lists assumptions following expert consultation with CHAI, for the level of effort needed to engage providers and achieve increased uptake (subject to uncertainty analysis). If a cost was associated with more than one category, the cost was subdivided across the relevant categories based on estimations from CHAI on the percentage of time spent on each category (online supplemental table S15). Next, we calculated the per-unit cost of each cost item (online supplemental tables S16, S17). For example, to calculate the cost of field officers to retain one provider per year, we multiplied the total cost of field officers in 2019, by the percentage of time spent by field officers on provider retention in 2019, divided by the number of active providers in 2019.

In practice, increasing the coverage of intervention activities is achieved through field officer visits, to engage directly with the providers. Finally, we also considered costs of routine TB services provided by the public sector, including the cost of diagnosis and the cost of treating DS-TB and DR-TB (online supplemental table S5).

We calculated the incremental cost-effectiveness ratio (ICER) by dividing the incremental cost of increasing the coverage of PPSA services by the number of TB cases averted between 2020 and 2035, relative to a baseline of indefinite continuation of current levels of routine TB and JEET PPSA services.

## Sensitivity analyses

We conducted various sensitivity analyses to explore the impact of certain model assumptions on model results. First, we conducted multivariate sensitivity analysis using partial rank correlation coefficient to assess which model parameters are most influential for model outputs.

We also conducted sensitivity analyses on provider behaviour. In the main analysis, we accounted for the fact that active providers may revert to an inactive state, and we assumed that the behaviour of newly active providers is the same as pre-existing active behaviours. First, we considered an optimistic PPSA scale-up baseline that assumes active providers remain active once activated. Second, we assumed that newly active providers have 25% and 50% less uptake than pre-existing active providers (ie, using Ahmedabad as an example, 9% of pre-existing active providers use CB-NAAT 100% of the time; we modelled scenarios where this proportion is 4.5% and 2.3%). We did not differentiate extrapulmonary and pulmonary TB in the model; however, CB-NAAT is unable to diagnose TB among patients with exclusively extrapulmonary disease. Thus, high levels of extrapulmonary TB can reduce the incremental benefit of CB-NAAT testing. In Ahmedabad and Delhi, it is estimated that 32% and 38% of patients diagnosed by JEET PPSA services between 2019 and 2022 had extrapulmonary TB, respectively. To account for this in a simple way, we reduced the proportion of patients with TB presenting to care that are correctly diagnosed by CB-NAAT by 50%. For the purpose of sensitivity analysis, this reflects an extreme scenario where 50% of patients who present to care have extrapulmonary TB.

Finally, we examined the impact of varying maintenance costs by ±20% on the ICER of the different interventions.

We defined maintenance costs as costs that are associated with maintaining good engagement and uptake among active providers (ie, maintaining notifications, FDC use and CB-NAAT use).

### Patient and public involvement

Patients or the public were not involved in the design, or conduct, or reporting, or dissemination plans of our research.

## RESULTS

Online supplemental figures S4, S5 show results of model calibration, displaying the model fit against each of the calibration targets listed in table 1.

Figure 2 shows model projections for the impact on TB incidence between 2017 and 2035 in Ahmedabad and Delhi across varying levels of PPSA coverage, with cumulative impact summarised in table 4. Relative to a comparator where current TB services are maintained, increasing the four PPSA activities simultaneously to their feasible maximum coverage would avert 5.1% (95% credible interval (CrI) 4.4% to 6.2%) of cumulative TB cases between 2020 and 2035, whereas increasing them to their theoretical maximum would avert 7.4% (95% CrI 6.2% to 9.2%) of cumulative TB cases between 2020 and 2035 in Ahmedabad. In Delhi, 5.4% (95% CrI 4.4% to 6.9%) and 6.7% (95% CrI 5.3% to 8.9%) of cumulative TB cases would be averted under feasible and theoretical maximum coverages, respectively. There would be a larger impact on DR-TB cases and deaths averted as three of the four PPSA activities involved increasing the use of CB-NAAT, thus increasing the early detection of

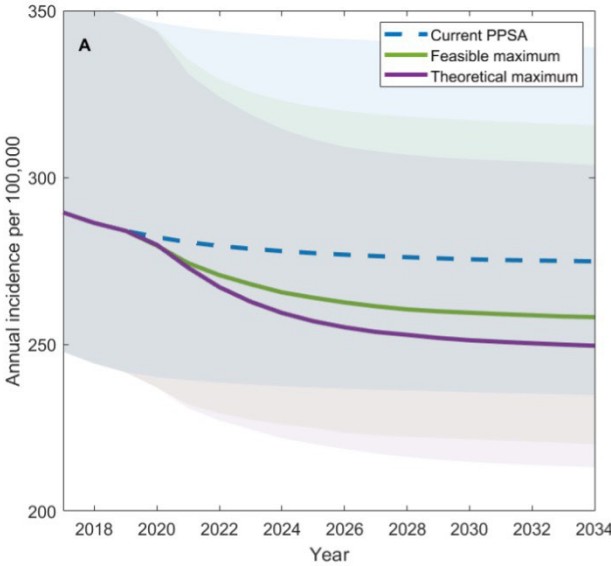
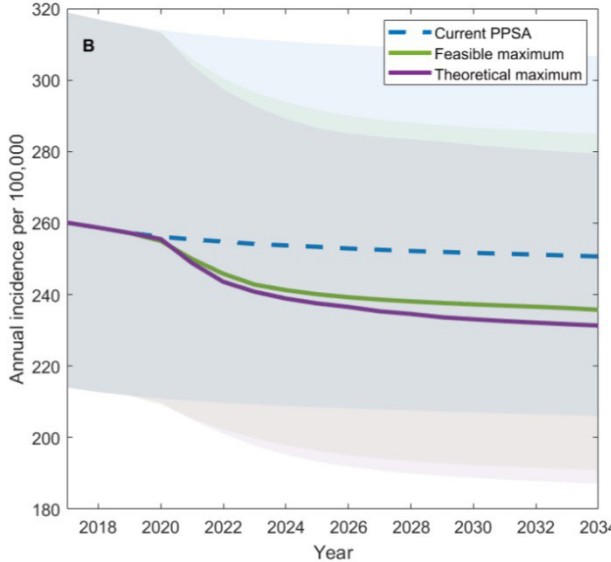

**Figure 2** Model projections for the impact of increasing all PPSA service provisions to their feasible or theoretical maximum has on TB incidence in Ahmedabad and Delhi between 2020 and 2035. (A, B) The projected trajectory of TB incidence in Ahmedabad and Delhi, respectively, assuming current PPSA scale up between 2017 and 2020, and its maintenance indefinitely (blue-dashed line), assuming that PPSA service provisions are increased to their feasible or theoretical maximum from 2020 to 2021 and maintained indefinitely (green and purple line, respectively). Shaded areas show 95% credible intervals. PPSA, public–private support agency; TB, tuberculosis.

**Table 4** Projected epidemiological impact

| Per cent reduction in cumulative TB burden, 2020–2035 | Ahmedabad | Delhi |
|---|---|---|
| Feasible maximum | | |
| TB incidence | 5.1% (4.4–6.2) | 5.4% (4.4–6.9) |
| TB deaths | 4.1% (2.8–5.8) | 6.4% (4.9–8.2) |
| DR-TB incidence | 8.0% (6.2–9.7) | 14% (11–17) |
| DR-TB deaths averted | 20% (18–24) | 32% (27–37) |
| Theoretical maximum | | |
| TB incidence | 7.4% (6.2–9.2) | 6.7% (5.3–8.9) |
| TB deaths | 7.9% (5.7–11) | 8.7% (6.8–11.3) |
| DR-TB incidence | 14% (11–16) | 18% (15–22) |
| DR-TB deaths averted | 35% (30–39) | 41% (35–47) |

Impact is measured as the cumulative impact on TB cases and deaths in Ahmedabad and Delhi between 2020 and 2035 relative to a baseline of current PPSA scale up.
DR-TB, drug-resistant tuberculosis; PPSA, public–private support agency.

rifampicin resistance. For example, assuming PPSA activities are increased to their theoretical maximum coverage 14% (95% CrI 11% to 16%) and 35% (95% CrI 30% to 39%) of cumulative DR-TB cases and deaths would be averted between 2020 and 2035, respectively. This impact increases to 18% (95% CrI 15% to 22%) and 41% (95% CrI 35% to 47%) DR-TB cases and deaths averted in Delhi, respectively, due to the lower current levels of CB-NAAT uptake in Delhi compared with Ahmedabad.

In Ahmedabad, the interventions with the lowest ICERS were provider activation and promoting FDC use,

costing an estimated US$18 000 (US$12 000–US$29 000) and US$36 000 (US$26 000–US$56 000) per TB case averted between 2020 and 2035, assuming the interventions were scaled up to their feasible maximum. Despite their low ICERs, these interventions were also the least epidemiologically impactful (online supplemental table S18, figure 3. The interventions with the highest ICERS were those promoting CB-NAAT, costing US$120 000 (US$96 000–US$180 000) and US$46 000 (US$37 000–US$68 000) per TB case averted depending on whether CB-NAAT use was promoted among occasional users and

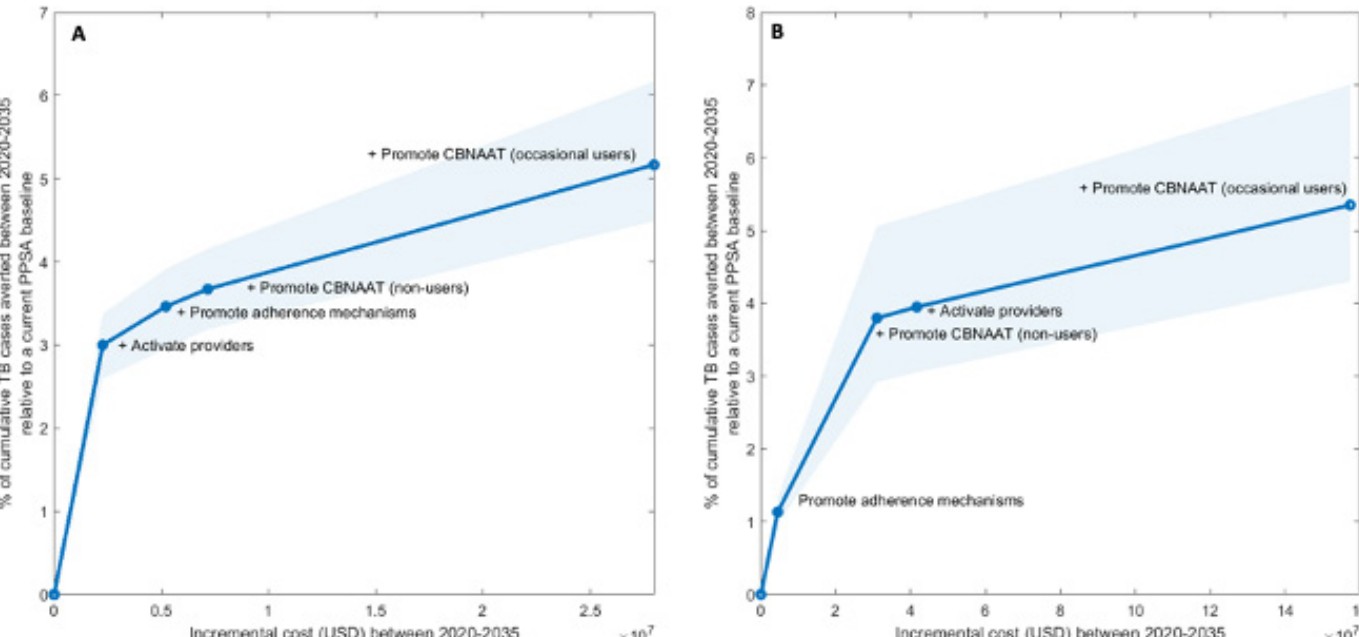

**Figure 3** Expansion pathway of PPSA service provisions in Ahmedabad (A) and Delhi (B). The first point on the graph represents the intervention that had the lowest ICER on its own when scaled up to its feasible maximum coverage; the second point represents the additional PPSA, in combination with the first point, had the lowest ICER and etc, until all interventions are included. Shaded area shows the 95% credible interval. See online supplemental tables S18, S19 for further details. CB-NAAT, cartridge-based nucleic acid amplification testing; ICER, incremental cost-effectiveness ratio; PPSA, public–private support agency; TB, tuberculosis.

non-users, respectively. For Delhi, promoting FDC use also had the lowest ICER costing an estimated US$170 000 (US$110 000–US$310 000) per TB case averted between 2020 and 2035 (online supplemental table S19, figure 3. However, promoting the use of CB-NAAT among non-users and provider activation had similar ICERs at US$210 000 (US$150 000–US$320 000) and US$280 000 (US$200 000–US$400 000) per TB case averted, respectively. Similar to Ahmedabad, promoting the use of CB-NAAT among occasional users had the highest ICER at US$780 000 (US$500 000–US$900 000) per TB case averted. Figure 3 summarises the order in which the different interventions should be prioritised in Ahmedabad and Delhi, and online supplemental tables S20, S21 provide a breakdown of the total costs.

Online supplemental figures S6–S8 show the results of sensitivity analyses. Alternate assumptions for the proportion of active providers that remain consistently active and the proportion of patients with TB with extrapulmonary TB reduced the epidemiological impact of increasing PPSA service provisions (online supplemental figure S6). The qualitative findings shown in figure 3 were unchanged by alternative assumptions on the uptake of CB-NAAT and treatment adherence among newly recruited providers (online supplemental table S18). However, assumptions relating to maintenance costs (the cost associated with maintaining good behaviour among active providers) showed some influence on the prioritisation of interventions. For Ahmedabad, decreasing maintenance cost by 20% resulted in the promotion of government supplied FDCs being the priority intervention (online supplemental table S22). On the other hand, increasing the proportion of active providers remained the individual intervention with the lowest ICER if maintenance cost increased by 20%. For Delhi, increasing the proportion of FDC users remained the intervention with the lowest ICER regardless of a 20% increase or decrease in maintenance cost (online supplemental table S23).

Finally, we conducted multivariate sensitivity analysis to identify the parameters that are most influential for model outputs. Model projections for TB cases averted in Ahmedabad were influenced most strongly by the following three parameters: the rate of relapse following self-cure, the rate of stabilisation of the risk of relapse following treatment and the rate of repeat care-seeking (online supplemental figure S7). For Delhi, the three most influential parameters were also the rate of relapse following self-cure and the rate of stabilisation of the risk of relapse following treatment, as well as the rate of progression to active disease from latency (online supplemental figure S8).

## DISCUSSION

Our analysis focuses on settings that have already seen an introduction of JEET services and addresses how these services should be expanded in future, to achieve reductions in TB burden in the most cost-effective way. Given

the wide variation in epidemiology and health services across different JEET locations, it is critical for any such analysis to be context-specific, taking account of local data. Our work illustrates how mathematical models can be helpful in doing so, by offering a tool for analysis of programmatic data. Focusing on Ahmedabad and Delhi, two cities with different coverage levels of PPSA services, our analysis suggests that JEET operations should prioritise working with already-engaged providers to encourage them to participate more actively in Ahmedabad and increasing the use of government supplied FDCs among already-active providers in Delhi. Similar modelling can be applied to identify priorities for future operations in other settings.

Previous modelling work estimated the potential epidemiological impact of scaling up PPSA operations in Mumbai and Patna, the cities where the PPSA pilots were first implemented.[10] Subsequent analysis used a relatively simple costing model to illustrate how PPSA scale up would be cost-effective in both settings.[7] These findings echoed earlier analysis that similarly showed how public–private mix in India would be cost-effective.[31] However, all these analyses were performed prospectively, and thus, we had to make assumptions about the effectiveness of different interventions, in cities where PPSA operations did not already exist at scale. Now that JEET has been operating at scale in India, there is detailed programmatic data available for the costs, effectiveness and real-world coverage of different interventions: our analysis thus adds to previous work by analysing this data in a systematic way, to identify operational priorities for future expansion of PPSA activities, in given cities.

Overall, the interventions with the highest ICERs are those that involve CB-NAAT. Although these interventions are epidemiologically impactful, they are costly, due to the high cost of a CB-NAAT test; the cost of testing symptomatic non-TB infected care-seekers; and the cost of engaging with the active providers to increase their use of CB-NAAT. The cost data used in this analysis suggest that reducing the effort needed to convince private providers to increase CB-NAAT use will be key in decreasing the overall cost. A recent study found that larger providers may be more willing to change their behaviour and adopt the use of CB-NAAT compared with smaller providers; thus, targeting these larger providers may provide a more cost-efficient solution and help bring down the cost of increasing the use of CB-NAAT among private providers.[8] On the other hand, promoting uptake of FDCs—although having modest impact—also consistently showed the lowest estimated ICERs, because of its low cost.

The epidemiological impact is likely to be reduced if there is a high rate of attrition from an active to an inactive status among private providers. There are several reasons behind this attrition: first, some providers may not have a sufficient load of patients with TB every month and thus may notify infrequently depending on when they see a new patient with TB; second, some providers, especially

those handling a smaller volume of patients with TB, are seen less regularly by the field team and are therefore less likely to notify; third, some providers remain dissatisfied with the services offered by JEET, and therefore, intentionally stop notifying patients; fourth, providers may be wary of the scrutiny they receive once they become active and involved with the programme. Understanding the reasons behind this behaviour will be important to ensure engagement efforts do not go to waste.

Limitations of this work can be divided into two categories: transmission model limitations and economic limitations. Owing to a lack of city-specific data, we drew from prevalence and infection surveys from Chennai, another large Indian city. This analysis focuses on urban settings and does not address rural settings, which may have different care-seeking pathways, different rates of TB infection and different provider behaviour.[3 19] While JEET activities have so far focused on urban India, addressing needs for private sector engagement in rural settings is an important area for future work. Another area for future work will be to extend this analysis to other cities where PPSA activities have been ongoing. With over 480 such cities to draw from, any future analysis could prioritise cities that capture the geographical and cultural variations across the country. On the cost side, we had to make certain assumptions to fill data gaps. For example, there are no systematic data for the number of visits needed to change the behaviour of a provider; this parameter was estimated by the CHAI field team as a form of expert opinion. Another assumption we made is that the field team visits providers resistant to engagement efforts for the same amount of time as non-resistant providers; in reality, after a couple of visits, field officers know which providers are resistant, and thus, may choose instead to focus their efforts on non-resistant providers. Similarly, we do not consider the fact that once a provider has been visited by field staff a certain number of times, any subsequent visits are likely to be ineffective; for example, data from the field team suggest that the increase in a provider's uptake of CB-NAAT becomes negligible after approximately the 25th visit by a field officer. Finally, while our work concentrates on PPSA activities, there are other important interventions not incorporated in the analysis, for example, Nikshay Poshan Yojana[32] an initiative to provide patients with TB with financial assistance towards nutritional support. While such measures are likely to have had important benefits to patients with TB, they are equally available to those reported from both public and private sectors: thus, they may not substantially change our model projections for future PPSA priorities. Nonetheless, another important area for future work is to incorporate data for these and other initiatives, to achieve more refined estimates for the potential impact of future PPSA activities.

In summary, as India accelerates efforts for drastically reducing TB burden, engaging effectively with the private sector will play a critical role. Modelling analysis can play a useful role in supporting these efforts, not only in informing strategic planning, but also in addressing operational priorities in different settings. Identifying the most cost-efficient ways of achieving PPSA impact will be critical in benefiting patients with TB, in the future of India's TB response.

**Contributors** NA conceptualised and supervised the study, and acted as guarantor. All authors contributed to the methodology. SR conducted the formal analysis. NA and SR wrote the original draft. AP, RS and AS contributed to data curation and reviewed the draft.

**Funding** SR was supported by the Wellcome Trust, grant number 203839/A/16/Z.

**Competing interests** None declared.

**Patient and public involvement** Patients and/or the public were not involved in the design, or conduct, or reporting, or dissemination plans of this research.

**Patient consent for publication** Not applicable.

**Provenance and peer review** Not commissioned; externally peer reviewed.

**Data availability statement** All data relevant to the study are included in the article or uploaded as online supplemental information.

**ORCID iDs**
Saskia Ricks http://orcid.org/0000-0001-7994-6111
Ridhima Sodhi http://orcid.org/0000-0003-2401-1065
Nimalan Arinaminpathy http://orcid.org/0000-0001-9502-6876

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
