## [Reviewer comments · BMJ Open]

ARTICLE DETAILS

TITLE (PROVISIONAL)	Operational priorities for engaging with India's private healthcare sector for the control of tuberculosis: a modelling study
AUTHORS	Ricks, Saskia; Singh, Ananya; Sodhi, Ridhima; Pal, Arnab; Arinaminpathy, Nimalan

VERSION 1 – REVIEW

REVIEWER	Mehta, Kedar GMERS Medical College Gotri Vadodara, Community Medicine Department
REVIEW RETURNED	25-Dec-2022

GENERAL COMMENTS	Congratulations to the team for this novel topic of PPSA interventions modelling for TB care. Some minor concerns to be addressed are as below: 1. It would be better if there is detailed explanation of selection of Ahmedabad and Delhi out of 485 cities where JEET intervention has been going. I would rather suggest to go for analysis zone wise like west, east, north, south - so that all geographic and cultural variations of our country could be taken care of.2. Kindly justify the results and how did you control the effect of other confounding factors like other private sector interventions by other NGOs, Direct money transfer to patients under Nikshay Poshan Yojana, etc...in your modelling.3. The paper is focused on 'provider recruitment, uptake of Xpert and uptake of adherence support mechanisms' - however I would suggest that it would be more interesting if you could add about PPSA effect on 'treatment outcome' also,
---

REVIEWER	Ejalu, David Makerere University College of Health Sciences, School of Public Health
REVIEW RETURNED	24-Feb-2023

GENERAL COMMENTS	This is a very pertinent study to the fight against TB. well written and informative.
---

REVIEWER	Workie, Addisalem Metu University
REVIEW RETURNED	24-Feb-2023

GENERAL COMMENTS	Title: Operational priorities for engaging with India's private healthcare sector for the control of tuberculosis: a modelling study I would like to appreciate the hard work for operationalize priorities for engaging with private healthcare sector to control tuberculosis.
---

	However, addressing the following comments might increase the quality of the overall manuscript.  1. Please provide the magnitude of Tuberculosis case from global to local. 2. What are the possible causes of tuberculosis? Please write causes in the introduction section clearly. 3. Please any prior experience of intervention to control the tuberculosis? Especially private health sectors' experience? 4. Any challenge they faced? 5. Explore the rationality of the study. 6. Little reference. 7. Keep the idea flow consistence while doing all these. 8. Please write methods section of the study in shortly and clearly? 9. What is the outcome of interest? 10. Avoid figure citation at the discussion section. 11. Please support your finding with other similar studies. 12. Please make sure all grammars issue in the manuscript.
--	---

VERSION 1 – AUTHOR RESPONSE

Reviewer: 1

1. It would be better if there is detailed explanation of selection of Ahmedabad and Delhi out of 485 cities where JEET intervention has been going. I would rather suggest to go for analysis zone wise like west, east, north, south - so that all geographic and cultural variations of our country could be taken care of.

We have added to the text to give some more context on these two cities, as well as to explain that they were selected because of their contrasting levels of PPSA services. Thus, these cities offer helpful examples of how future priorities for PPSA expansion may differ, depending on the current state of PPSA services. Please see p.4, line 3, “We focused on two urban settings, ...” and the text that follows.

We appreciate the suggestion of geographically diverse cities, but hope the reviewer will appreciate the value of selecting cities based on their current levels of PPSA services instead. Nonetheless, we have also added to the discussion, to mention the need for future work to address different geographies. Please see p.13, line 28 “Another area for future work...”

2. Kindly justify the results and how did you control the effect of other confounding factors like other private sector interventions by other NGOs, Direct money transfer to patients under Nikshay Poshan Yojana, etc...in your modelling.

We did not address for these confounding factors in our analysis, as the impact of these additional factors are hard to quantify, and their city-level data was unavailable to us. We have added to the discussion to elaborate more on these important interventions. Please see p.14 line 2, “Finally, while our work concentrates on PPSA activities...” and the text that follows.

3. The paper is focused on 'provider recruitment, uptake of Xpert and uptake of adherence support mechanisms' - however I would suggest that it would be more interesting if you could add about PPSA effect on 'treatment outcome' also

We would like to clarify that the model indeed incorporates treatment outcomes: the purpose of the adherence support intervention is to improve treatment completion amongst patients treated by private providers, and we used data reported to CHAI to quantify the extent of these improvements (please see “Proportion of TB patients that do not complete treatment”, bottom of Table S4). Please also see the main text describing treatment outcomes on p.7, line 17 the paragraph beginning “We considered 4 categories of active providers...”

Reviewer: 2
No comments

Reviewer: 3

1. Please provide the magnitude of Tuberculosis case from global to local.

We have added information on global TB cases and deaths, as well as the proportion of global TB burden accounted for by India. Please see page 3 lines 2-4,

2. What are the possible causes of tuberculosis? Please write causes in the introduction section clearly.

As this manuscript is not intended as a review of tuberculosis, we have refrained from going into too much background on the disease, that may detract from the overall flow. Nonetheless, we have added a reference that is relevant for readers wanting to know more about the general context. Please see p.3, first line of the introduction.

3. Please any prior experience of intervention to control the tuberculosis? Especially private health sectors' experience?

We have added to the introduction to offer more background. Again, to maintain proper flow and focus on the topic of the study, we have focused on background relevant to private sector engagement in India. Please see p.3 line 9, “Early efforts to engage effectively with the private sector...” and the text that follows.

4. Any challenge they faced?

Please see text referenced in response to comment 3: here we briefly mention key challenges that were faced by early public-private mix initiatives in India.

5. Explore the rationality of the study.

To our understanding, this comment is about the need to clarify the rationale for the study. We have added text to the introduction to make this clearer. Please see p.3, line 10, “Despite these achievements, there remains much ground to be covered...”, and the text that follows. We hope this addresses the reviewer’s comment.

6. Little reference.

We have added more references throughout the introduction, including along with the new text in response to the comments above. In the methods, we have made it more explicit which data was provided by CHAI.

7. Keep the idea flow consistence while doing all these.

We hope the reviewer will agree that the manuscript has retained a logical flow with all the revisions now incorporated.

8. Please write methods section of the study in shortly and clearly?

We have organised the manuscript so that the main text provides an overview necessary for understanding the model, while the supporting information provides all technical details that are necessary to replicate the findings. We feel that shortening the Methods section too much could compromise clarity for readers who are interested in how the model is structured. Nonetheless, in order to improve the flow, we have moved some material to the supporting information, mainly the extensive tables of parameters. We hope the reviewer will agree that the overall text is now more accessible, while maintaining its flow.

9. What is the outcome of interest?

The aim of the study was to assess the cost-effectiveness of the different interventions offered by JEET. This is already mentioned in the introduction and discussion, but we have now added to the text to emphasise this overarching aim. Please see:

p.3, line 35, "...to identify the most cost-effective strategies for future PPSA expansion."
 p.12, line 18, "...to achieve reductions in TB burden in the most cost-effective way."

10. Avoid figure citation at the discussion section.

We have removed figure citations from the discussion section.

11. Please support your finding with other similar studies.

To our knowledge, ours is the first analysis that identifies priorities for PPSA expansion based on measures that would reduce burden in the most cost-effective way possible. Thus there are no existing studies that are directly comparable. Nonetheless, we have cited a recent study in India that addressed overall cost-effectiveness of PPSAs (relative to a comparator of no PPSA), as well as an earlier study that first showed how cost-effective public-private mix could be in India. Please see p.12 line 30, "Subsequent analysis used a relatively simple costing model..."

12. Please make sure all grammars issue in the manuscript.

We have removed any grammatical issues throughout the manuscript.

VERSION 2 – REVIEW

REVIEWER	Ejalu, David Makerere University College of Health Sciences, School of Public Health		
REVIEW RETURNED	16-Jul-2023		
GENERAL COMMENTS	Operational priorities for engaging with India’s private healthcare sector for the control of tuberculosis: a modelling study		
	Comments		
	Section	Line	Comment

	Abstract	Line 2: Objective	The information provided is more of background information. I suggest the to clearly state the objective of the modelling study.
		Line 30: Strength and limitations	Modelling studies are based mostly Assumptions, the use of assumptions as a limitation does not weigh much, unless there is a specific assumption that might have affected the results, its better to state that specific assumption.
	Introduction		Informative
	Methods	Page 4, Line 10	What was the justification of choosing adults above 15 years, state
		Page 4, line 19	non-TB symptomatic population was calibrated to a 10% prevalence of TB amongst those presenting for care, we do not see the consideration of the non-TB population in the care cascade
			Other methods sections were all described and well elaborate.
	Results		The results adequately address the research questions However, a table showing the variations of the costs and cost effectiveness of each intervention per state would have made the results clearer.
	Discussion	Page 12; line 17-26	In simple terms, do these findings simply mean that the cities should continue doing what they are already doing? What are the major implications of these findings, especially in regards to what is being currently implemented?
		Page 13; line 7-8	“Reducing the effort needed to convince private providers to increase CB-NAAT use will be key in decreasing the overall cost” Table 3 already shows that the No. of active providers that always use CB-NAAT is already low in both cities. Although its not clear whether this low number is mostly contributed by private providers.
		Page 13; line 14-17	Are these reason based assumptions or the authors or there is some evidence to allude to these?
Summary		Great summary	

REVIEWER	Workie, Addisalem Metu University
REVIEW RETURNED	28-May-2023
GENERAL COMMENTS	As I have seen in the revised manuscript, the authors have not made substantial revisions to the original manuscript, and do not properly address comments. However, I couldn't see any new references added in the references section.

VERSION 2 – AUTHOR RESPONSE

“The information provided is more of background information. I suggest the authors to clearly state the objective of the modelling study.”

Please note that the objective of the study was already mentioned, towards the end of the introduction. Nonetheless, we have adjusted the text, to state the research question in a more focused way. Please see p.3, line 30: “In particular, in a given city...” and the text that follows.

Similarly, we have also adjusted the abstract to make the objective clearer. Please see p.2, line 2: “Objectives: to estimate...” and the text that follows.

“Modelling studies are based mostly on assumptions, the use of assumptions as a limitation does not weigh much, unless there is a specific assumption that might have affected the results, it’s better to state that specific assumption.”

We have made the final bullet point under the “Strengths and limitations” section more specific. Please see p.7, line 32, “Due to data gaps in healthcare provider behaviour...” and the text that follows.

“What was the justification of choosing adults above 15 years, state”.

Choosing adults above the age of 15 years old is consistent with the age categories we have calibrated our model to. We have deleted the “>15 years old” from page 4 line 10 and have moved it to the “Data sources” section on page 4 line 27 where we discuss the studies, we have calibrated our model to.

“non-TB symptomatic population was calibrated to a 10% prevalence of TB amongst those presenting for care, we do not see the consideration of the non-TB population in the care cascade”.

Please note that the non-TB symptomatic population has no influence at all on the TB transmission dynamics, nor on the TB care cascade: its purpose is solely to allow quantification of the number of diagnostic tests being used, as well as the number of unnecessary TB treatments (for the purpose of costing). Although this was mentioned in the main text, we have now included additional text to emphasise this point. Please see p.4 line 18, “This population has no influence on the ‘main’ model of TB transmission dynamics, ...” and the text that follows.

We have also added text to highlight further information available in the supporting information. Please see p.5 line 23, “The supplementary information also provides further details on the non-TB symptomatic population care cascade.”

“The results adequately address the research questions. However, a table showing the variations of the costs and cost effectiveness of each intervention per state would have made the results clearer.”

Please note that tables S15-23 onwards in the supplementary information provides a breakdown of costs for Ahmedabad and Delhi, as well as the incremental cost of each intervention in each state. These tables are referenced throughout on page 11 line 10 onwards.

“In simple terms, do these findings simply mean that the cities should continue doing what they are already doing?” and “What are the major implications of these findings, especially in regards to what is being currently implemented?”

Every PPSA offers a range of different services: improved diagnostics, adherence support mechanisms, efforts to recruit new private providers, etc. In all settings, there are inevitable gaps in the implementation of these different services, and these gaps will vary from city to city, depending on the local health system and epidemiological context.

Therefore, rather than simply continuing what cities already doing, our analysis helps to identify which gaps are most important to address in any given setting. Unfortunately, it is not possible to make a single recommendation for all cities. This is because such recommendations need to be context-specific, taking account of local factors such as epidemiology, existing PPSA implementation, etc. Our work shows how modelling can be used to take account of all of these factors. For example in Ahmedabad, we identify that the PPSA should prioritise provider recruitment, whereas in Delhi, the PPSA should prioritise increasing the uptake of adherence support mechanisms. Similarly, this kind of modelling can be applied to other cities as well, to identify their priorities.

We have amended the discussion to make this point more clear, and to highlight that Delhi and Ahmedabad are only examples. Please see p.12, line 21, “Given the wide variation in epidemiology and health services...” and the text that follows. Please also see p.4, line 3, “As two examples, we focused on the urban settings, ...”

“Reducing the effort needed to convince private providers to increase CB-NAAT use will be key in decreasing the overall cost”. Table 3 already shows that the No. of active providers that always use CB-NAAT is already low in both cities. Although it’s not clear whether this low number is mostly contributed by private providers. Are these reason-based assumptions or the authors or there is some evidence to allude to these?

The figures in table 3 are exclusively for private providers that JEET engage with. To avoid confusion, we have amended the table caption to refer to ‘active private providers’. Please note that this table shows operational data, not assumptions. We have also added to the caption to explain this. Finally, the quoted statement is based on the economic data provided, and in particular on the cost involved in encouraging uptake of CB-NAAT amongst private providers. We have added a few words to make it clearer: please see p.13 line 11, “The cost data used in this analysis...”

VERSION 3 – REVIEW

REVIEWER	Ejalu, David Makerere University College of Health Sciences, School of Public Health
REVIEW RETURNED	21-Dec-2023
GENERAL COMMENTS	This is a very important paper. To the best of my knowledge, the authors have addressed all the comments raised in the previous review. Just like any other peer reviewed paper, there are limitations, but these cannot deter the publication. Work well done by the authors.